

# Diffusion coefficients of organic molecules in sucrose-water solutions and comparison with Stokes-Einstein predictions

Yuri Chenyakin, Dagny A. Ullmann, Erin Evoy, Lindsay Renbaum-Wolff[†], Saeid Kamal, Allan K. Bertram

5  Department of Chemistry, University of British Columbia, Vancouver, BC, V6T 1Z1, Canada
† now affiliated with Aerodyne Research, Inc, Billerica, MA 01821 and Boston College, Chestnut Hill, MA 02467

*Correspondence to:* A. K. Bertram (bertram@chem.ubc.ca) and S. Kamal (skamal@chem.ubc.ca)

## Abstract

Diffusion coefficients of organic species in secondary organic aerosol (SOA) particles are needed to predict the growth and reactivity of these particles in the atmosphere. Previously, viscosity measurements along with the Stokes-Einstein relation

have been used to estimate diffusion rates of organics within SOA particles or proxies of SOA particles. To test the Stokes-Einstein relation, we have measured the diffusion coefficients of three fluorescent organic dyes (fluorescein, Rhodamine 6G and calcein) within sucrose-water solutions with varying water activity. Sucrose-water solutions are used as a proxy for SOA material found in the atmosphere. Diffusion coefficients were measured using fluorescence recovery after photobleaching. For the three dyes studied the diffusion coefficients varies by 5-7 orders of magnitude as the water activity

varied from 0.38 to 0.88, illustrating the sensitivity of the diffusion coefficients to the water content in the matrix. At the lowest water activity studied (0.38) the average diffusion coefficients were $1.8 \times 10^{-5}$, $1.6 \times 10^{-6}$ and $7.6 \times 10^{-6}$ µm$^2$ s$^{-1}$ for fluorescein, Rhodamine 6G and calcein, respectively. The measured diffusion coefficients were compared with predictions made using literature viscosities and the Stokes-Einstein relation. We found that at a water activity $\geq 0.6$ (which corresponds to a viscosity $\leq 360$ Pa s and $T_g/T \leq 0.81$) predicted diffusion rates agreed with measured diffusion rates within the

experimental uncertainty. ($T_g$ represents the glass transition temperature and T is the temperature of the measurements). When the water activity was 0.38 (which corresponds to a viscosity of $3.3 \times 10^6$ Pa s and a $T_g/T$ of 0.94) the Stokes-Einstein relation under-predicted the diffusion coefficients of fluorescein, Rhodamine 6G and calcein by a factor of 95 (minimum 7 and maximum of 980), a factor of 17 (minimum 1 and maximum 165) and a factor of 56 (minimum 7 and maximum 465), respectively. The observed disagreement is significantly smaller than the disagreement observed when comparing measured

and predicted diffusion coefficients of water in sucrose-water mixtures.





## 1 Introduction

Large quantities of volatile organic compounds such as isoprene, α-pinene, and toluene are emitted into the atmosphere annually. Subsequently, these molecules are oxidized in the atmosphere to form semivolatile organic compounds, which can condense to the particle phase and form secondary organic aerosol (SOA). Although the exact chemical composition of SOA is not known, the average oxygen-to-carbon elemental ratio of SOA ranges from approximately 0.2 to 1.0 (Aiken et al., 2008; Chen et al., 2009; DeCarlo et al., 2008; Hawkins et al., 2010; Heald et al., 2010; Jimenez et al., 2009; Ng et al., 2010; Takahama et al., 2011). Due to the hygroscopic nature of SOA (Hildebrandt Ruiz et al., 2015; Massoli et al., 2010), an important component of SOA particles is water. To emphasize this point, in the following we will refer to these particles as SOA-water particles. As the relative humidity (RH) varies in the atmosphere from low values to 100%, the water content (or water activity, $a_w$) of the SOA-water particles will also vary, from low values to high values to maintain equilibrium with the gas-phase.

In order to predict properties of SOA-water particles information on the diffusion rates of water, oxidants, and organic molecules within these particles is needed. For example, information on the diffusion of water within SOA-water particles is needed for predicting their cloud condensation abilities and ice nucleating abilities (Adler et al., 2013; Berkemeier et al., 2014; Bones et al., 2012; Lienhard et al., 2015; Price et al., 2015; Schill et al., 2014; Wang et al., 2012; Wilson et al., 2012). Information on the diffusion rates of oxidants and organic molecules are needed for predicting the heterogeneous chemistry and photochemistry of these particles (Davies and Wilson, 2015; Gržinić et al., 2015; Hinks et al., 2016; Houle et al., 2015; Kuwata and Martin, 2012; Li et al., 2015; Lignell et al., 2014; Shiraiwa et al., 2011; Wang et al., 2015; Wong et al., 2015; Zhou et al., 2012). Diffusion rates of organic molecules within SOA-water particles is also needed for predicting growth rates and size distributions of these particles, as well as the long range transport of polycyclic aromatic hydrocarbons in the atmosphere (Virtanen et al., 2010; Shiraiwa and Seinfeld, 2012; Shiraiwa et al. 2013, Zaveri et al., 2014; Zelenyuk et al., 2012). Due to the importance of diffusion within SOA-water particles, many studies have recently focused on this topic (e.g. (Abramson et al., 2013; Bateman et al., 2016; Kidd et al., 2014; Lu et al., 2014; Marshall et al., 2016; Pajunoja et al., 2014, 2015; Perraud et al., 2012; Robinson et al., 2013; Saleh et al., 2013; Yatavelli et al., 2014; Zhang et al., 2015)).

In the following we focus on the diffusion of organics within SOA-water particles. To predict diffusion rates of organics within SOA-water particles, some researchers, including ourselves, have used viscosities of SOA-water particles or proxies of SOA-water particles together with the Stokes-Einstein relation (Booth et al., 2014; Hosny et al., 2013; Koop et al., 2011; Power et al., 2013; Renbaum-Wolff et al., 2013a, 2013b; Shiraiwa et al., 2011; Song et al., 2015, 2016). Given below (Eq. 1) is the Stokes-Einstein relation for the case of no slip at the surface of the diffusing species within a fluid:

$$D = \frac{kT}{6\pi\eta R_H}, \tag{1}$$



where $D$ is the diffusion coefficient, $k$ is the Boltzmann constant, $T$ is temperature in Kelvin, $\eta$ is the dynamic viscosity and $R_H$ is the hydrodynamic radius of the diffusing species. Studies are needed to quantify when the Stokes-Einstein relation does and does not provide accurate estimates of the diffusion within SOA-water particles and proxies of SOA-water particles under atmospherically relevant conditions.

Most previous studies that have tested the validity of the Stokes-Einstein equation have used single-component (and often nonpolar) matrices (Blackburn et al., 1994, 1996; Chang et al., 1994; Cicerone et al., 1995; Ehlich and Sillescu, 1990; Fujara et al., 1992; Heuberger and Sillescu, 1996; Rossler and Sokolov, 1996; Rossler, 1990). There have also been a few studies (partially motivated by applications in food science) that have tested the validity of the Stokes-Einstein equation for predicting diffusion of organics in organic-water matrices (Champion et al., 1997; Corti et al., 2008a, 2008b; Rampp et al., 2000). This work has shown that the Stokes-Einstein relation under predicts the diffusion coefficient of organics in organic-water matrices close to the glass transition temperature, although the temperature range over which break down occurs is not completely resolved.

Herein, we expand on the previous measurements of diffusion of organics in organic-water matrices. Specifically, we measured the diffusion coefficients of three fluorescent organic dyes within sucrose-water mixtures as a function of $a_w$, and we have compared the measurements with predictions using the Stokes-Einstein relation. Sucrose-water mixtures were used as the matrix in these studies for several reasons: 1) the viscosities of sucrose-water mixtures has been reported for a wide range of atmospherically relevant $a_w$-values, 2) the oxygen-to-carbon ratio of sucrose (0.92) is in the range of O:C values observed in oxidized atmospheric particles and 3) the room temperature viscosities of sucrose-water solutions are similar to the room temperature viscosities of some types of SOA-water particles (e.g. compare viscosities of sucrose-water solutions from Power et al. (2013) with viscosities of SOA-water particles generated from toluene photooxidation (Song et al., 2016), isoprene photooxiation (Song et al., 2015) and $\alpha$-pinene ozonolysis (Grayson et al., 2016)). The organic dyes chosen for these experiments were fluorescein, Rhodamine 6G and calcein. Shown in Fig. 1 are the structures of these dyes and listed in Table 1 are their molecular weights (MW) and hydrodynamic radii ($R_H$).

## 2 Experimental

Rectangular area fluorescence recovery after photobleaching (rFRAP) (Deschout et al., 2010) was used to measure diffusion coefficients of the fluorescent organic dyes in sucrose-water mixtures. For these experiments, thin films (30–50 μm thick) of sucrose, water, and trace amounts of fluorescent dye (< 0.5 wt. %) were required. In Section 2.1, the methods used to generate the thin films are discussed and in Section 2.2, the rFRAP technique is described.



## 2.1 Preparation of thin films containing sucrose, water and trace amounts of fluorescent dye

The concentrations of sucrose in the thin films studied ranged from 50 to 93 wt. % sucrose, which corresponds to $a_w$-values ranging from 0.93 and 0.38. To prepare thin films that were subsaturated with respect to crystalline sucrose (i.e. < 67 wt. % sucrose and $a_w$ > 0.84), solutions of sucrose, water, and trace amounts of dye were prepared gravimetrically; then, a 0.5 µL droplet of the solution was pipetted onto a siliconized hydrophobic glass slide (Hampton Research), resulting in a droplet of approximately 350 µm in radius on the slide; next, a second hydrophobic slide was placed on top of the first slide containing the droplet. The two slides were pushed together, sandwiching the droplet and forming a thin film between the two slides with a thickness of 30–50 µm, determined by an aluminum spacer (Fig. 2). High-vacuum grease around the perimeter of the slides provided a seal.

To prepare thin films that were supersaturated with respect to crystalline sucrose (i.e. concentrations > 67 wt. % sucrose and $a_w$ < 0.84), the following method was used: first, a solution containing 60 wt. % sucrose in water and trace amounts of dye were prepared gravimetrically; then, the solution was passed through a 0.02 µm filter (Whatman™) to eliminate impurities (e.g. dust), and a droplet of the prepared solution was placed on a siliconized hydrophobic slide (Hampton Research); next, the hydrophobic slide containing the droplet was placed inside a flow cell or sealed glass container with a controlled relative humidity (RH). In cases where a flow cell was used, the RH was controlled using a humidified flow of $N_2$ gas (Bodsworth et al., 2010; Koop et al., 2000; Pant et al., 2004). In cases where a sealed glass container was used, the RH was set by placing supersaturated inorganic salt solutions with known water vapour partial pressures (Greenspan, 1977) within the sealed glass containers. The relative humidity was measured with a hygrometer with an uncertainty of ± 2.5 %. The slide holding the droplet was left inside the flow cell or sealed glass containers for an extended period of time (see Supplement, section S1 and Table S1 to S3) to allow the droplet enough time to come to equilibrium with the surrounding RH. Once equilibrium is reached, the activity of water in the droplet and the gas-phase are equal and $a_w$ can be calculated from RH. The wt. % of sucrose in the droplet was then calculated using the relationship between $a_w$ and wt % sucrose given by Eq. 2 (Zobrist et al., 2011):

$$a_w\,(T,w) = \frac{1+aw}{1+bw+cw^2} + (T - T^\theta)(dw + ew^2 + fw^3 + gw^4), \tag{2}$$

where $T$ is the temperature of the experiments (294.5 +/- 1.0 K), $T^\theta$ is a reference temperature of 298.15 K, $w$ is the sucrose weight fraction, $a=-1$, $b=-0.99721$, $c=0.13599$, $d=0.001688$, $e=-0.005151$, $f=0.009607$ and $g=-0.006142$. After the droplet on the slide was conditioned to a known RH, the droplet was sandwiched between another siliconized hydrophobic slide producing a film of approximately 30-50 µm in thickness, determined by an aluminum spacer (Fig. 2). As mentioned above, high-vacuum grease around the perimeter of the slides provided a seal. The process of sandwiching the droplet was carried out within a Glove Bag™ (Glas-Col), which was inflated with humidified $N_2$ gas. The humidity within the Glove Bag™ was set to the same RH as used to condition the droplet, to prevent the droplet from being exposed to an unknown and





uncontrolled RH. Once the thin films were generated and sealed with high-vacuum grease, they were also kept over saturated inorganic salt solutions (in a sealed container) with RH values equal to the RH used to condition the droplets.

For the experiments where thin films were supersaturated with respect to crystalline sucrose, crystallization was not
5   observed in most cases. This was likely because the solutions were first passed through a 0.02 µm filter to remove any heterogeneous nuclei that could initiate crystallization and the glass slides used to make the thin films were coated with a hydrophobic material, which reduces significantly the ability of these surfaces to promote heterogeneous nucleation (Bodsworth et al., 2010; Pant et al., 2004, 2006; Price et al., 2014; Wheeler and Bertram, 2012). In the few cases where crystallization was observed, the films were not used in the rFRAP experiments.

The concentrations of the dyes in the thin films were approximately 0.8 mM, 0.4 mM, and 0.3 mM for fluorescein, Rhodamine 6G, and calcein, respectively. To prepare thin films containing these dyes, fluorescein disodium salt (Sigma-Aldrich), rhodamine 6G chloride (Acros Organics) and calcein (Sigma-Aldrich) were used.  To dissolve calcein in sucrose-water solutions, small amounts (< 0.5 wt. %) of NaOH were required. Concentrations of the dyes were chosen so that 1) the
concentrations were small enough to not significantly influence the viscosity of the sucrose-water solutions, 2) the fluorescence signal was large enough to detect in the rFRAP experiments, and 3) the intensity of the fluorescence signal was linear with concentration of the fluorescent dyes for the range used in the rFRAP experiments. In a separate set of experiments, the intensity of the fluorescence signal as a function of the dye concentration in sucrose-water films was measured (see Supplement, Section S2 and Fig. S1-S3). The intensity of the fluorescence signal was found to be linear for
the concentrations of dyes used in our experiments.

## 2.2 rFRAP technique

The technique of fluorescence recovery after photobleaching (FRAP) is often utilized in the biological and materials science communities to measure diffusion coefficients in biological materials, single cells, and organic polymers (see refs. Braeckmans et al., 2003, 2007; Hatzigrigoriou et al., 2011; Seksek et al., 1997; Smith et al., 1981and references therein).
The rFRAP technique is a recently developed version of FRAP (Deschout et al., 2010).  In the rFRAP experiments, a small volume of the thin film was photobleached with a confocal laser scanning microscope, decreasing the fluorescence signal in the photobleached volume. After photobleaching, the fluorescence in this volume was monitored with the same confocal microscope for an extended period of time. Due to molecular diffusion of organic fluorescent probe molecules, the fluorescence in the photobleached volume recovered, and from the time-dependent recovery of the fluorescence signal, the
diffusion coefficient was determined. Additional details are given below.

For the experiments performed using fluorescein and calcein dyes, the rFRAP experiments were performed on a Leica TCS SP5 II confocal laser scanning microscope with a 10x, 0.4 numerical aperture (NA) objective and a pinhole setting of 53µm.





Photobleaching was performed using a 488 nm Ar laser set at 1.18 mW, and after photobleaching images were acquired with the same laser line at 2.2 µW. Experiments were performed using Leica FRAP Wizard software, using the "Zoom-In" bleach mode.

For the experiments performed using Rhodamine 6G, the rFRAP experiments were performed on a Zeiss Axio Observer LSM 510 MP laser scanning microscope with a 10x, 0.3 NA objective and a pinhole setting of 80 µm. Photobleaching was performed using a 543 nm HeNe laser set at 330 µW. After photobleaching, images were acquired with the same laser line at 4.08 µW laser intensity. Experiments were performed using the Zen 2008 software, using the "Zoom-In" bleach mode.

In all experiments, the exposure time used for photobleaching was chosen such that it resulted in approximately 30 % of the fluorescent molecules being photobleached in the region of interest (ROI) as suggested by Deschout et al. (2010). Deschout et al. (2010) previously showed that diffusion coefficients measured with rFRAP were independent of the extent of photobleaching up to a depletion of 50 % of the fluorescent signal in the ROI.

The geometry of the photobleached region was rectangular, with a length $l_x$ and a width $l_y$. Bleached areas ranged from 5 x 5 µm$^2$ to 36 × 36 µm$^2$, depending on the diffusion rates. Smaller photobleached regions were used in cases with slow diffusion rates to shorten the fluorescence recovery time. The specific bleach sizes used in the experiments are indicated in Tables S1–S3. In a separate set of experiments, we measured the diffusion coefficient of calcein in a 72 wt. % sucrose thin film as a function of bleach area. The results show that the diffusion coefficients varied by less than the uncertainty in the 20 measurements when the bleach size was varied from 1 x 1 to 50 × 50 µm$^2$ (Fig. S4), consistent with previous rFRAP studies (Deschout et al., 2010).

**2.3 Extraction of diffusion coefficients from rFRAP data**

Shown in Fig. 3 are examples of images recorded during a rFRAP experiment. Panel A shows an image of the film prior to photobleaching, and panels B-F show images after photobleaching. All the images after photobleaching are normalized using 25 an image recorded prior to photobleaching or using an area in each image not influenced by photobleaching. To reduce noise, all images were converted from a resolution of 512 × 512 pixels to 128 × 128 pixels by averaging.

The images recorded during the rFRAP experiments (e.g. Fig. 3) represent fluorescence intensities as a function of position x and y for different times t after photobleaching. The mathematical description for fluorescence intensity as a function of x, y 30 and t after photobleaching a rectangular profile with a laser scanning confocal microscope was given by Deschout et al. (2010):





$$\frac{F(x,y,t)}{F_0(x,y)} = 1 - \frac{K_0}{4}\left(erf\left(\frac{x+\frac{l_x}{2}}{\sqrt{w(t)}}\right) - erf\left(\frac{x-\frac{l_x}{2}}{\sqrt{w(t)}}\right)\right) \bullet \left(erf\left(\frac{y+\frac{l_y}{2}}{\sqrt{w(t)}}\right) - erf\left(\frac{y-\frac{l_y}{2}}{\sqrt{w(t)}}\right)\right), \tag{3}$$

where $F(x,y,t)$ represents the fluorescence intensity at positions x and y and at a time $t$ after photobleaching, $F_0(x,y)$ is the fluorescence intensity at positions x and y prior to photobleaching, $K_0$ is related to the fraction of molecules photobleached in the rectangle and $l_x$ and $l_y$ are the lengths of the photobleached rectangle in the x and y directions, respectively. The parameter $w$ is described by the following equation:

$$w(t) = r^2 + 4Dt, \tag{4}$$

where $r$ is the resolution parameter of the microscope and $D$ is the diffusion coefficient of the dye. Eq. (3) was derived with the assumption that the degree of photobleaching is independent of the z-direction (i.e. the depth in the thin film), which is a reasonable approximation in our experiments due to the use of thin films (30-50 µm) and the use a 10x objective lens with a low numerical aperture, which gives a near cylindrical geometry over a distance of 30-50 µm in the z-direction (Deschout et al., 2010).

Through a fitting procedure, Eq. (3) was used to extract values of $w(t)$ from the fluorescence images recorded after photobleaching. In the fitting procedure, $K_0$, $w(t)$ and the location of the center of the photobleached region were left as free parameters, as well as an additional normalization factor, which usually returned a value close to 1 since the images were normalized prior to fitting. After values of $w(t)$ were determined from each of the fluorescence images, $w(t)$ was plotted versus $t$ such as in Fig. 4. A straight line was then fit to this data, and the diffusion coefficient was determined from slope of the line and Eq. (4).

For each concentration of sucrose and for each organic dye, the diffusion coefficient was determined 9 times (3 different thin films were used and 3 measurements were carried out on each thin film).

## 3 Results and Discussion

### 3.1 Diffusion coefficients of the three fluorescent organic dyes in sucrose-water solutions

Shown in Fig. 5 are diffusion coefficients for fluorescein in sucrose-water solutions. Several different x-axes (wt.% sucrose, $a_w$, $T_g/T$, and viscosity) are included to put the results in context. The water activities for samples subsaturated with respect to sucrose were calculated using Eq. (2). $T_g$ and T are the glass transition temperature and temperature of the matrix, respectively. $T_g$ was calculated from wt. % sucrose using the relationship between $T_g$ and wt. % sucrose given in Champion et al. (1997). Viscosity was calculated from $a_w$ using viscosity data (Migliori et al., 2007; Power et al., 2013; Quintas et al., 2006; Telis et al., 2007) parameterized as a function $a_w$.





Fig. 5 illustrates that the diffusion coefficient of fluorescein in sucrose-water solutions is strongly dependent on $a_w$, with the diffusion coefficient varying by almost 6 orders of magnitude as $a_w$ varied from 0.38 to 0.88. This strong dependence of the diffusion coefficient on $a_w$ is because water acts as a plastizer in sucrose-water mixtures – as the water content in the matrix increases the viscosity of the matrix decreases (Power et al., 2013). At the lowest $a_w$ studied, the average diffusion

coefficient of fluorescein was $1.8 \times 10^{-5}$ $\mu m^2$ $s^{-1}$.

To test the Stokes-Einstein relation, in Fig. 5 the measured diffusion coefficients for fluorescein are compared with diffusion coefficients calculated with the Stokes-Einstein relation and previous viscosity measurements of sucrose-water solutions (Migliori et al., 2007; Power et al., 2013; Quintas et al., 2006; Telis et al., 2007). To calculate the diffusion coefficients, a

hydrodynamic radius of 5.02 Å was used for fluorescein based on measurements of fluorescein diffusion coefficients in water (Mustafa et al., 1993). At $a_w \geq 0.6$ (which corresponds to a $T_g/T \leq 0.81$ and a viscosity $\leq 360$ Pa s) the measured diffusion coefficients are consistent with the predicted diffusion coefficients. At a water activity of 0.38 (which corresponds to a $T_g/T$ value of 0.94 and a viscosity of approximately $3.3 \times 10^6$ Pa s) the Stokes-Einstein equation under-predict the diffusion coefficient by a factor of approximately 95 (minimum factor of 7 and maximum factor of 980 if the uncertainties in

the measured diffusion coefficients and the predicted diffusion coefficients are considered).

Shown in Figs. 6 and 7 are diffusion coefficients of Rhodamine 6G and calcein in sucrose-water solutions. Diffusion coefficients of these two dyes also depended strongly on $a_w$. For Rhodamine 6G, the diffusion coefficient appears to vary by more than 7 orders of magnitude as $a_w$ varies from 0.38 to 0.88. For calcein, the diffusion coefficient varied approximately 5

orders of magnitude as $a_w$ was varied from 0.38 to 0.88. At the lowest $a_w$ studied (0.38) the average diffusion coefficient for Rhodamine 6G and calcein were $1.6 \times 10^{-6}$ and $7.6 \times 10^{-6}$ $\mu m^2$ $s^{-1}$, respectively.

Also included in Figs. 6 and 7 are diffusion coefficients calculated using the Stokes-Einstein relation and viscosities of sucrose-water solutions reported in the literature (Migliori et al., 2007; Power et al., 2013; Quintas et al., 2006; Telis et al.,

2007). When calculating diffusion coefficients using the Stokes-Einstein equation, hydrodynamic radii of 5.89 Å and 7.4 Å were used for Rhodamine 6G and calcein, respectively, based on measured diffusion coefficients of these dyes in water (Müller and Loman, 2008; Tamba et al., 2010). Figs. 6 and 7 show that, similar to fluorescein, the measured diffusion coefficients are consistent with the predicted diffusion coefficients at $a_w \geq 0.6$ (which corresponds to $T_g/T \leq 0.81$ and a viscosity $\leq 360$ Pa s). On the other hand, at a water activity of 0.38 (which corresponds to a $T_g/T$ value of 0.94 and a

viscosity of approximately $3.3 \times 10^6$ Pa s), the Stokes-Einstein equation appear to under-predicted the diffusion coefficients: For Rhodamine 6G, the measured diffusion coefficient is greater than the predicted diffusion coefficient by a factor of approximately 17 (minimum factor of 1 and maximum factor of 165 if the uncertainties in the measured diffusion coefficients and the predicted diffusion coefficients are considered). For calcein, the measured diffusion coefficient is greater



than the predicted diffusion coefficient by approximately 56 (minimum factor of 7 and maximum factor of 465 if the uncertainties in the measured diffusion coefficients and the predicted diffusion coefficients are considered).

### 3.2 Comparison with previous measurements of organics or organometallics in sucrose-water matrices.

In Table 2, we summarize previous studies that tested the Stokes-Einstein relation using organics or organometallics in sucrose-water mixtures. Champion et al. (1997) measured diffusion coefficients of fluorescein in sucrose-water solutions at temperatures ranging from 20 °C to -15 °C, and Corti et al. (2008) measured diffusion coefficients of fluorescein in sucrose-water solutions at approximately 20 °C. The results from Champion et al. (1997) indicate that the Stokes-Einstein relation under predicted diffusion coefficients for $T_g/T \gtrsim 0.9$, while good agreement is observed at smaller $T_g/T$ values. The results from Corti et al. (2008) show disagreement between measured and predicted diffusion coefficients for $T_g/T \gtrsim 0.7$ and good agreement at smaller $T_g/T$ values. Longinotti and Corti (2007) measured the diffusion of ferrocene methanol in sucrose-water solutions. Their results indicate that the Stokes-Einstein relation under predicts diffusion coefficients for $T_g/T \gtrsim 0.8$, while good agreement is observed at smaller $T_g/T$ values.

In our studies with fluorescein, Rhodamine 6G and calcein, breakdown of the Stokes-Einstein relation is observed at a $T_g/T$ value of approximately 0.93 and no indication of breakdown is apparent at a $T_g/T$ value of approximately 0.81. At a $T_g/T$ value of 0.87 there is some indication of breakdown in our studies since the measured average diffusion coefficient for fluorescein and Rhodamine 6G is outside the 95% prediction intervals. These observations are consistent with the results from Champion et al., and the $T_g/T$ values where we observed breakdown is only slightly higher than the values based on Corti et al. (2008) and Longinotti and Corti (2007).

### 3.3 Comparison with the diffusion of water in sucrose-water solutions

Compared to the fluorescent organic dyes studied here, larger disagreement has been observed between measured and predicted diffusion coefficients for water in sucrose-water mixtures (Power et al., 2013; Price et al., 2014). To illustrate this point, in Fig. 8 the diffusion coefficients of water in sucrose-water solutions measured by Price et al. (2014) are shown and compared with predicted diffusion coefficients for water in sucrose-water solutions based on the Stokes-Einstein relation and viscosity measurements. The measurements by Price et al. are in good agreement with other measurements at $a_w \geq 0.3$ (Davies and Wilson, 2016; Price et al., 2014; Rampp et al., 2000; Zobrist et al., 2011). To predict the diffusion coefficients of water in Fig. 8, a hydrodynamic radius of 1.41 Å was used (Pang, 2014). Fig. 8 shows that even at a water activity of 0.6, the Stokes-Einstein relation under-predicts the diffusion coefficient by a factor between approximately 10 and 1000. At a water activity of 0.38, the Stokes-Einstein under-predicts the diffusion coefficient of water by a factor of approximately $10^3$ to $10^5$. For the case of small molecules like water, other relations besides the Stokes-Einstein relation may be needed (Essam, 1980; Marshall et al., 2016; Molinero et al., 2003; Murata et al., 1999).



## 4 Summary and Conclusions

Using rFRAP, we measured diffusion coefficients of three fluorescent organic dyes (fluorescein, Rhodamine 6G and calcein) in sucrose-water solutions for water activities ≥ 0.38 (which correspond to viscosities ≤ $3.3 \times 10^6$ Pa s and $T_g/T \leq 0.94$). The diffusion coefficients of the organic dyes depended strongly on the water activity, with the diffusion coefficients varying by approximately 5-7 orders of magnitude as $a_w$ varied from 0.38 to 0.88.

The measured diffusion coefficients were compared to diffusion coefficients calculated using the Stokes-Einstein relation and viscosities from the literature. For all three dyes studied, the Stokes-Einstein relation predicts diffusion coefficients in agreement with the measured diffusion coefficients when $a_w \geq 0.6$ or when the solution viscosity is ≤ 360 Pa s and $T_g/T \leq$ 0.81. In contrast, at $a_w = 0.38$ or when the solution viscosity equals $3.3 \times 10^6$ Pa s and $T_g/T = 0.94$, the Stokes-Einstein relation under-predicted the diffusion coefficients of fluorescein, Rhodamine 6G and calcein by a factor of 95 (minimum 7 and maximum of 980), a factor of 17 (minimum 1 and maximum 165) and a factor of 56 (minimum 7 and maximum 465), respectively.

The range of $T_g/T$ values over which we observed breakdown of the Stokes-Einstein relation is broadly consistent with previous measurements that tested the breakdown of the Stokes-Einstein relation using organics or organometallics in sucrose-water mixtures. Compared to the fluorescent organic dyes studied here, larger disagreement has been observed between measured and predicted diffusion coefficients of water in sucrose-water mixtures (Power et al., 2013; Price et al., 2014). At a water activity of 0.38, the Stokes-Einstein under-predicts the diffusion coefficient of water by a factor of approximately $10^3$ to $10^5$. The results presented here should be useful for developing corrections for the Stokes-Einstein equation and making estimations of diffusion rates of organic molecules in secondary organic aerosol particles found in the atmosphere.

## Acknowledgements

This work was carried out in the Laboratory for Advanced Spectroscopy and Imaging Research (LASIR) at the University of British Columbia, Vancouver and supported by funding from the Natural Sciences and Engineering Research Council of Canada and the Canadian Foundation for Innovation.



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

30





**Table 1.** The molecular weights (MW) and hydrodynamic radii ($R_H$) of the fluorescent organic dyes used in this work.

| Organic dye | MW (g/mol) | $R_H$ (Å) |
|---|---|---|
| Fluorescein | 332 | 5.02 (Mustafa et al., 1993) |
| Rhodamine 6G | 443 | 5.89 (Müller and Loman, 2008) |
| Calcein | 622 | 7.4  (Tamba et al., 2010) |

**Table 2.** Summary of results from previous studies that tested the breakdown of the Stokes-Einstein relation using organics or organometallics in sucrose-water mixtures.

| Matrix | Diffusing molecule | $T_g/T$ where breakdown is clearly discernable | Reference |
|---|---|---|---|
| Sucrose-water | fluorescein | $\gtrsim 0.9$ | (Champion et al., 1997) |
| Sucrose-water | fluorescein | $\gtrsim 0.68$-$0.78$ | (Corti et al., 2008a) |
| Sucrose-water | ferrocene methanol | $\gtrsim 0.8$ | (Longinotti and Corti, 2007) |



**Figures**

**Figure 1.** Molecular structures (neutral forms) of the three fluorescent organic dyes used in this work: fluorescein (A), Rhodamine 6G (B) and calcein (C).




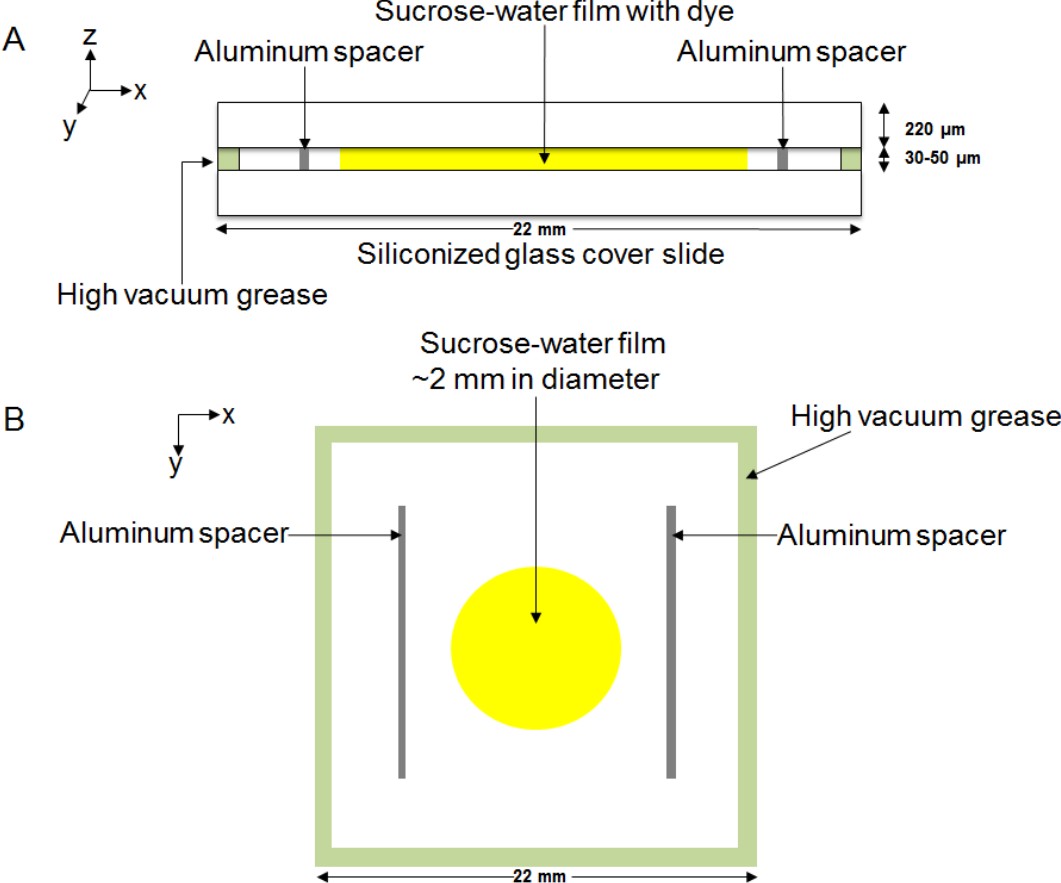

**Figure 2.** Side view (A) and top view (B) of a thin film containing sucrose, water, and a fluorescent dye sandwiched between two hydrophobic glass slides as prepared for use in rFRAP experiments.





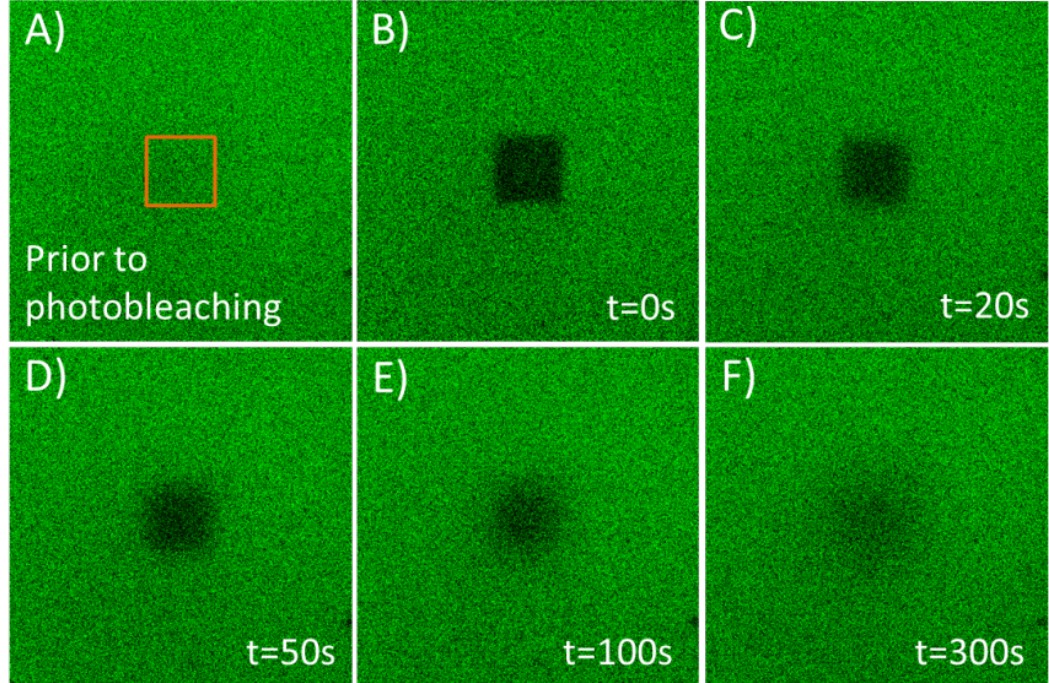

**Figure 3.** Images recorded during an rFRAP experiment using a thin film composed of 66 wt. % sucrose solution ($a_w$ = 0.85) and trace amounts of rhodamine 6G (0.4 mM). (A)Image recorded before photobleaching, (B) image recorded immediately after photobleaching a $36 \times 36$ $\mu m^2$ area  and  (C-F) are images recorded at time (t) of 20, 50, 100 and 300 seconds after photobleaching, respectively. The orange square in panel (A) represents the $36 \times 36$ $\mu m^2$ area selected for photobleaching.





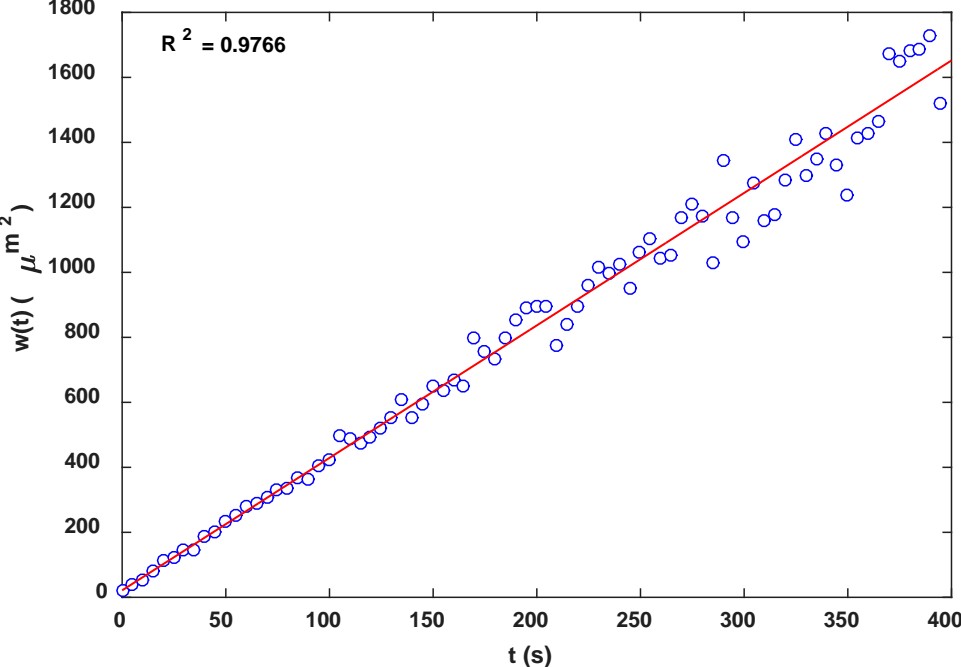

**Figure 4.** Plot of *w(t)* versus time for rhodamine 6G in a 66 wt. % sucrose solution ($a_w = 0.85$). The red line is a linear fit to the data. The diffusion coefficient was determined from the slope of the line.





**Figure 5.** A comparison of measured diffusion coefficients of fluorescein in sucrose-water films from this work (red stars) with predicted diffusion coefficients based on measured viscosities of sucrose-water solutions and the Stokes-Einstein equation from Power et al. (2013) (blue squares), Migliori et al. (2007) (blue crosses), Telis et al. (2007) (blue circles) and Quintas et al. (2006) (blue triangles). The x-error bars for this work correspond to the uncertainty in the determination of $a_w$. The y-errors for this work correspond to 95% confidence intervals. Several different x-axes (wt.% sucrose, $a_w$, $T_g/T$, and viscosity) are included to help put the results in context. T represents the temperature of the experiment (294.5 K) and $T_g$ represent the glass-transition temperature of sucrose-water solutions.



**Figure 6.** A comparison of measured diffusion coefficients of rhodamine 6G in sucrose-water films from this work (red stars) with predicted diffusion coefficients based on measured viscosities of sucrose-water solutions and the Stokes-Einstein equation from Power et al. (2013) (blue squares), Migliori et al. (2007) (blue crosses), Telis et al. (2007) (blue circles) and Quintas et al. (2006) (blue triangles). The x-error bars for this work correspond to the uncertainty in the determination of $a_w$. The y-errors for this work correspond to 95% confidence intervals. Several different x-axes (wt.% sucrose, $a_w$, $T_g/T$, and viscosity) are included to help put the results in context. T represents the temperature of the experiment (294.5 K) and $T_g$ represent the glass-transition temperature of sucrose-water solutions.





**Figure 7.** Comparison of measured diffusion coefficients of calcein in sucrose-water films from this work (red stars) with predicted diffusion coefficients based on measured viscosities of sucrose-water solutions and the Stokes-Einstein equation from Power et al. (2013) (blue squares), Migliori et al. (2007) (blue crosses), Telis et al. (2007) (blue circles) and Quintas et al. (2006) (blue triangles). The x-error bars for this work correspond to the uncertainty in the determination of $a_w$. The y-errors for this work correspond to 95% confidence intervals. Several different x-axes (wt.% sucrose, $a_w$, $T_g/T$, and viscosity) are included to help put the results in context. T represents the temperature of the experiment (294.5 K) and $T_g$ represent the glass-transition temperature of sucrose-water solutions.



**Figure 8.** A comparison of measured diffusion coefficients of water in sucrose-water films from Price et al. (2014) (red stars) with predicted diffusion coefficients based on measured viscosities of sucrose-water solutions and the Stokes-Einstein equation from Power et al. (2013) (blue squares) Migliori et al. (2007) (blue crosses), Telis et al. (2007) (blue circles) and Quintas et al. (2006) (blue triangles). Several different x-axes (wt. % sucrose, $a_w$, $T_g/T$, and viscosity) are included to help put the results in context. T represents the temperature of the experiment (294.5 K) and $T_g$ represent the glass-transition temperature of sucrose-water solutions.