# Peer review of "Diffusion coefficients of organic molecules in sucrose-water solutions and comparison with Stokes-Einstein predictions"

_Atmospheric Chemistry and Physics, 2016_

## Referee Comment (RC1) · Anonymous Referee #1 · 12 Sep 2016

In light of evidence for particle-phase diffusion imposing a kinetic limitation on gas/particle partitioning and other processes in the atmosphere, studies of diffusion are required to improve estimates or diffusivity. The aim of this paper is to improve our understanding of the accuracy of the Stokes-Einstein equation for converting measured viscosities to diffusion coefficients, and to present new measurements of diffusion coefficients in a system of atmospheric relevance. The method is well described and shows efforts have been made to minimise error, and references are given to its previous application. Results are presented very well (the multiple x-axes in figs. 5-8 are very useful) and the discussion is suitably informative and concise. Results are compared to those from previous studies in a useful and insightful discussion. Overall, the

study is suitable for the journal and is of a good quality.

The following are recommended minor revisions:

i) Clarification of what, if any, method was used to ensure that the weight fraction of sucrose in subsaturated (with regard to sucrose) samples was maintained between gravimetric preparation and sealing within slides. It is clear that a controlled RH environment was used to do this for supersaturated samples, but it is not stated how it was done for subsaturated samples. If no method was employed could it be shown, either through measurement or theory, that weight fraction is expected to be maintained between preparation and sealing?

ii) To show, either through measurement or theory, that there is negligible change to sample temperature as a result of laser exposure (since this would affect diffusivity).

iii) To make clear the source of plotted uncertainties. If these are from measurement repeats, this should be stated.

iv) It seems that uncertainty may be introduced by relative humidity measurements and scatter in plots used to derive the diffusion coefficient (e.g. fig. 4). If these are not factored into the plotted error bars, what uncertainty do they introduce?

v) Would be useful for comparison (perhaps in the supplement) to have on one plot the diffusion coefficient vs. water activity relationship for all three dyes and for water from the Price et al. 2014 study.

Technical point: i) Is the unit for y-axis on figure 4 right? Looks odd.

---

## Referee Comment (RC2) · Anonymous Referee #2 · 23 Sep 2016

The manuscript describes a novel method, applied to a problem in atmospheric aerosol science for the first time, to measure the diffusion of organic components with deposited droplets of aqueous solutions of sucrose at varying water activity and, thus, viscosity. Not only is the technique a significant new contribution in its own right to study aerosols, but the measurements that result are extremely useful in contributing to the continuing debate about the diffusion constants of organic species in viscous matrices in secondary organic aerosol. The manuscript is well-written and should be accepted for publication once the authors have responded to the following comments.

- To be consistent with all previous work and avoid confusion, I recommend that all units for diffusion constants be either cm2 s-1 or m2 s-1 throughout the manuscript.

- The uncertainty in RH of +/- 2.5 % is quite large. Given the steepness in the viscosity vs water activity dependence, significant error must be incurred from this. For conditioning as supersaturated solutions, how long was required for conditioning?

- Page 7: "...independent of the z-direction (i.e. the depth in the thin film), which is a reasonable approximation in our experiments due to the use of thin films (30-50 $\mu$m) and the use a 10x objective lens with a low numerical aperture, which gives a near cylindrical geometry over a distance of 30-50 $\mu$m in the z-direct." It is surprising to me that a 10x objective gives such a large Rayleigh range in the beam waist. I would have expected the beam to diverge more quickly than this beyond the focal waist and for the shape to not be of simple cylindrical geometry. Can the authors back up this claim with some actual calculations for their particular system?

- Photobleaching recovery: Following on from the last point about the geometry of the bleached volume, it must be presumed that there is no recovery following photobleaching apart from that due to diffusion of organic species back into the detection window of this assumed geometry. It would be helpful to confirm this by carrying out photobleaching recovery experiments at very low relative humidity in sucrose where the diffusion is most likely entirely quenched. Have the authors done this?

- Hydrodynamic radius of fluorescein: How confident can the authors be that the hydrodynamic radius is independent of water activity? This would seem to be quite crucial when considering the plausibility of the Stokes-Einstein equation for reproducing the diffusion constants from measured viscosities. Given that many of the estimated diffusion constants overlap with the measured diffusion constants, it seems possible to me that Stokes-Einstein could even be considered to be a good representation of the diffusion constants of the organic dyes, given all of the uncertainties/errors involved (including the uncertainties in RH and the uncertainties in the viscosity measurements of sucrose solutions).

- Given the disparity in sizes of the different fluorescent probes, would the authors

expect any of them to be better represented by Stokes-Einstein behavior, once their sizes are compared with that of sucrose?

---

## Author Comment (AC1) · 1 Dec 2016

Dr. David Topping
Co-Editor of Atmospheric Chemistry and Physics

Dear Dave,

Listed below are our responses to the comments from the reviewers of our manuscript. For clarity and visual distinction, the referee comments or questions are listed here in black and are preceded by bracketed, italicized numbers (e.g. *[1]*). Authors' responses are in red below each referee statement with matching numbers (e.g. *[A1]*). We thank the reviewers for carefully reading our manuscript and for their very helpful suggestions!

Sincerely,

Allan Bertram
Professor of Chemistry
University of British Columbia

Anonymous Referee #1

In light of evidence for particle-phase diffusion imposing a kinetic limitation on gas/particle partitioning and other processes in the atmosphere, studies of diffusion are required to improve estimates or diffusivity. The aim of this paper is to improve our understanding of the accuracy of the Stokes-Einstein equation for converting measured viscosities to diffusion coefficients, and to present new measurements of diffusion coefficients in a system of atmospheric relevance. The method is well described and shows efforts have been made to minimise error, and references are given to its previous application. Results are presented very well (the multiple x-axes in figs. 5-8 are very useful) and the discussion is suitably informative and concise. Results are compared to those from previous studies in a useful and insightful discussion. Overall, the study is suitable for the journal and is of a good quality. The following are recommended minor revisions:

*[1]* Clarification of what, if any, method was used to ensure that the weight fraction of sucrose in subsaturated (with regard to sucrose) samples was maintained between gravimetric preparation and sealing within slides. It is clear that a controlled RH environment was used to do this for supersaturated samples, but it is not stated how it was done for subsaturated samples. If no method was employed could it be shown, either through measurement or theory, that weight fraction is expected to be maintained between preparation and sealing?

*[A1]* To address the referee's question, we will remove from the paper all data corresponding to subsaturated (with regard to sucrose) samples. This only removes four data points from the paper and does not change the conclusions of the manuscript in any way.

*[2]* To show, either through measurement or theory, that there is negligible change to sample temperature as a result of laser exposure (since this would affect diffusivity).

*[A2]* To address the referees comment, we will measure diffusion coefficients as a function of laser intensity.

*[3]* To make clear the source of plotted uncertainties. If these are from measurement repeats, this should be stated.

*[A3]* For Fig. 5-7, the x-error bars for this work correspond to the uncertainty in the determination of $a_w$ from the hygrometer or from equation 2. The y-errors for this work correspond to 95% confidence intervals from measurement repeats. This will be made clear in the revised manuscript.

*[4]* It seems that uncertainty may be introduced by relative humidity measurements and scatter in plots used to derive the diffusion coefficient (e.g. fig. 4). If these are not factored into the plotted error bars, what uncertainty do they introduce?

*[A4]* For each concentration of sucrose and for each organic dye, the diffusion coefficient was determined 9 times (3 different thin films were used and 3 measurements were carried out on each thin film), and the y-errors for this work correspond to 95% confidence intervals from these repeats. These y-error bars should include the uncertainty from deriving the diffusion

coefficients (e.g. fig. 4). The x-error bars were introduced to account for the uncertainty of the relative humidity measurements. In the revised manuscript, we will make it more clear what the error bars represent in the figures.

[5] Would be useful for comparison (perhaps in the supplement) to have on one plot the diffusion coefficient vs. water activity relationship for all three dyes and for water from the Price et al. 2014 study.

[A5] We will add the plot suggested to the revised manuscript.

[6] Technical point: i) Is the unit for y-axis on figure 4 right? Looks odd.

[A6] Yes, the unit is correct. $w(t) = r^2 + 4Dt$, where $r$ is the resolution parameter of the microscope and $D$ is the diffusion coefficient of the dye. Hence, $w$ has units of $\mu m^2$.

Anonymous Referee #2

The manuscript describes a novel method, applied to a problem in atmospheric aerosol science for the first time, to measure the diffusion of organic components with deposited droplets of aqueous solutions of sucrose at varying water activity and, thus, viscosity. Not only is the technique a significant new contribution in its own right to study aerosols, but the measurements that result are extremely useful in contributing to the continuing debate about the diffusion constants of organic species in viscous matrices in secondary organic aerosol. The manuscript is well-written and should be accepted for publication once the authors have responded to the following comments.

[7] To be consistent with all previous work and avoid confusion, I recommend that all units for diffusion constants be either cm2 s-1 or m2 s-1 throughout the manuscript.

[A7] The units for diffusion constants will be changed to $cm^2 s^{-1}$ in the revised manuscript.

[8] The uncertainty in RH of +/- 2.5 % is quite large. Given the steepness in the viscosity vs water activity dependence, significant error must be incurred from this
. For conditioning as supersaturated solutions, how long was required for conditioning?

[A8] Conditioning times are reported in the supplemental. To make this clearer in the revised manuscript we will add the following text:

"Calculations of the time required for each droplet to come to equilibrium with the surrounding RH (i.e. conditioning time) is discussed in the Supplement, Section S1 and reported in Tables S1-S3. Conditioning times used in this work ranged from 30 min to 93 day."

[9] Page 7: "...independent of the z-direction (i.e. the depth in the thin film), which is a reasonable approximation in our experiments due to the use of thin films (30-50 µm) and the use

a 10x objective lens with a low numerical aperture, which gives a near cylindrical geometry over a distance of 30-50 µm in the z-direct." It is surprising to me that a 10x objective gives such a large Rayleigh range in the beam waist. I would have expected the beam to diverge more quickly than this beyond the focal waist and for the shape to not be of simple cylindrical geometry. Can the authors back up this claim with some actual calculations for their particular system?

*[A9]* Based on the referee's comments it is clear that our discussion on Page 7 has led to some confusion. The author is right that the geometry is not exactly a cylindrical geometry. Nevertheless, based on previous work by Deschout et al. it is close enough to a cylindrical geometry to give accurate diffusion coefficients when combined with Equation 3. To make this clear this section will be modified to the following:

"Although Eq. 3 was derived with the assumption that the degree of photobleaching is independent of the z-direction (i.e. the depth in the thin film), Deschout et al. have shown that Eq. (3) can be used to extract accurate diffusion coefficients when used with a 10x objective lens with a low numerical aperture (0.45) together with thin films (120 µm thick), since this combination gives close to a cylindrical photobleached geometry. In our work we used lower numerical apertures (0.3-0.4) and thinner films (30-50 µm) than Deschout et al."

*[10]* Photobleaching recovery: Following on from the last point about the geometry of the bleached volume, it must be presumed that there is no recovery following photobleaching apart from that due to diffusion of organic species back into the detection window of this assumed geometry. It would be helpful to confirm this by carrying out photobleaching recovery experiments at very low relative humidity in sucrose where the diffusion is most likely entirely quenched. Have the authors done this?

*[A10]* Besides diffusion, the only other mechanism of recovery that we can think of is reversible photobleaching. To address the referee's comments, we will carry out the following experiment: 1) we will prepare droplets with sizes between 10-80 µm in diameter containing sucrose, water and trace amounts of dye. 2) We will photobleached the dye uniformly throughout the droplet. 3) We will monitor the signal as a function of time after photobleaching. Since the photobleaching will be performed uniformly on the entire droplet the dye concentration should be uniform throughout the droplet after photobleaching, which eliminates the possibility of diffusion due to concentration gradients. This experiment will allow us to assess if reversible photobleaching is important in our FRAP experiments.

*[11]* Hydrodynamic radius of fluorescein: How confident can the authors be that the hydrodynamic radius is independent of water activity? This would seem to be quite crucial when considering the plausibility of the Stokes-Einstein equation for reproducing the diffusion constants from measured viscosities. Given that many of the estimated diffusion constants overlap with the measured diffusion constants, it seems possible to me that Stokes-Einstein could even be considered to be a good representation of the diffusion constants of the organic dyes, given all of the uncertainties/errors involved (including the uncertainties in RH and the uncertainties in the viscosity measurements of sucrose solutions).

*[A11]* The hydrodynamic radius of fluorescein could decrease as a function of water activity, and this decrease could explain some of the deviation from the Stokes-Einstein equation. However, the hydrodynamic radius is not expected to decrease by an order of magnitude when the water activity is varied from 0.6 to 0.38. To address the referee's comment this discussion will be added to the revised manuscript.

*[12]* Given the disparity in sizes of the different fluorescent probes, would the authors expect any of them to be better represented by Stokes-Einstein behavior, once their sizes are compared with that of sucrose?

*[A12]* The hydrodynamic radius of fluorescein, Rhodamine 6G, and calcein are 5.02, 5.89, and 7.4 Angstroms, respectively. The radius of sucrose is roughly 4.5 Angstroms based on the density of amorphous sucrose. Assuming break-down of the Stokes-Einstein equation only depends on the ratio of the radius of the fluorescent probe to the radius of the matrix molecules, we would expect the best agreement for calcein. Unfortunately, the uncertainties in our experiments are too large to test this relationship. We will add this discussion to the revised manuscript.

---

## Author Response (AR1)

Dr. David Topping
Co-Editor of Atmospheric Chemistry and Physics

Dear Dave,

Listed below are our responses to the comments from the reviewers of our manuscript. For clarity and visual distinction, the referee comments or questions are listed here in black and are preceded by bracketed, italicized numbers (e.g. *[1]*). Authors' responses are in red below each referee statement with matching numbers (e.g. *[A1]*). We thank the reviewers for carefully reading our manuscript and for their very helpful suggestions!

Sincerely,

Allan Bertram
Professor of Chemistry
University of British Columbia

Anonymous Referee #1

In light of evidence for particle-phase diffusion imposing a kinetic limitation on gas/particle partitioning and other processes in the atmosphere, studies of diffusion are required to improve estimates or diffusivity. The aim of this paper is to improve our understanding of the accuracy of the Stokes-Einstein equation for converting measured viscosities to diffusion coefficients, and to present new measurements of diffusion coefficients in a system of atmospheric relevance. The method is well described and shows efforts have been made to minimise error, and references are given to its previous application. Results are presented very well (the multiple x-axes in figs. 5-8 are very useful) and the discussion is suitably informative and concise. Results are compared to those from previous studies in a useful and insightful discussion. Overall, the study is suitable for the journal and is of a good quality. The following are recommended minor revisions:

*[1]* Clarification of what, if any, method was used to ensure that the weight fraction of sucrose in subsaturated (with regard to sucrose) samples was maintained between gravimetric preparation and sealing within slides. It is clear that a controlled RH environment was used to do this for supersaturated samples, but it is not stated how it was done for subsaturated samples. If no method was employed could it be shown, either through measurement or theory, that weight fraction is expected to be maintained between preparation and sealing?

*[A1]* To address the referee's question, we will remove from the paper all data corresponding to subsaturated (with regard to sucrose) samples. This only removes four data points from the paper and does not change the conclusions of the manuscript in any way.

*[2]* To show, either through measurement or theory, that there is negligible change to sample temperature as a result of laser exposure (since this would affect diffusivity).

*[A2]* Although there could be local heating during the photobleaching step, this is not expected to affect the measured diffusion coefficient since thermal diffusivity in the samples is orders of magnitude faster than molecular diffusivity. For example the thermal diffusivity of water is $\sim 1 \times 10^{-3}$ cm$^2$ s$^{-1}$ at room temperature, while molecular diffusion in our experiments is $\lesssim 1 \times 10^{-8}$ cm$^2$ s$^{-1}$. As a result any local heating during photobleaching will be dissipated to the surrounding environment on a time scale much shorter than the measurements of molecular diffusion. The measurements of diffusion coefficients as a function of bleach area support this conclusion. In these experiments the energy absorbed by the bleached region was varied by three orders of magnitude. Nevertheless the measured diffusion coefficient was found to be independent of the amount of energy absorbed by the bleached region. To address the referee's comments this discussion has been added to the manuscript (Section 2.2)

*[3]* To make clear the source of plotted uncertainties. If these are from measurement repeats, this should be stated.

*[A3]* For Fig. 5-7, the x-error bars for this work correspond to the uncertainty in the determination of $a_w$ from the hygrometer. The y-errors for this work correspond to 95% confidence intervals from measurement repeats. This has been made clear in the revised figure captions.

*[4]* It seems that uncertainty may be introduced by relative humidity measurements and scatter in plots used to derive the diffusion coefficient (e.g. fig. 4). If these are not factored into the plotted error bars, what uncertainty do they introduce?

*[A4]* For each concentration of sucrose and for each organic dye, the diffusion coefficient was determined at least times (3 different thin films were used and at least 3 measurements were carried out on each thin film), and the y-errors for this work correspond to 95% confidence intervals from these repeats. These y-error bars should include the uncertainty from deriving the diffusion coefficients (e.g. fig. 4). The x-error bars were introduced to account for the uncertainty of the relative humidity measurements. In the revised figure captions, we have tried to make it more clear what the error bars represent in the figures.

*[5]* Would be useful for comparison (perhaps in the supplement) to have on one plot the diffusion coefficient vs. water activity relationship for all three dyes and for water from the Price et al. 2014 study.

*[A5]* The plot suggested has been added to the revised manuscript (Fig. 9).

*[6]* Technical point: i) Is the unit for y-axis on figure 4 right? Looks odd.

*[A6]* Yes, the unit is correct. $w(t) = r^2 + 4Dt$, where $r$ is the resolution parameter of the microscope and $D$ is the diffusion coefficient of the dye. Hence, $w$ has units of $\mu m^2$.

Anonymous Referee #2

The manuscript describes a novel method, applied to a problem in atmospheric aerosol science for the first time, to measure the diffusion of organic components with deposited droplets of aqueous solutions of sucrose at varying water activity and, thus, viscosity. Not only is the technique a significant new contribution in its own right to study aerosols, but the measurements that result are extremely useful in contributing to the continuing debate about the diffusion constants of organic species in viscous matrices in secondary organic aerosol. The manuscript is well-written and should be accepted for publication once the authors have responded to the following comments.

*[7]* To be consistent with all previous work and avoid confusion, I recommend that all units for diffusion constants be either cm2 s-1 or m2 s-1 throughout the manuscript.

*[A7]* The units for diffusion constants have been changed to $cm^2 s^{-1}$ in the revised manuscript.

*[8]* The uncertainty in RH of +/- 2.5 % is quite large. Given the steepness in the viscosity vs water activity dependence, significant error must be incurred from this. For conditioning as supersaturated solutions, how long was required for conditioning?

*[A8]* Conditioning times are reported in the supplemental.  To make this clearer in the revised manuscript we have added the following text (Section 2.1):

"Calculations of the time required for each droplet to come to equilibrium with the surrounding RH (i.e. conditioning time) is discussed in the Supplement, Section S1 and reported in Tables S1-S3.  Conditioning times used in this work ranged from 30 min to 93 day."

*[9]* Page 7: "...independent of the z-direction (i.e. the depth in the thin film), which is a reasonable approximation in our experiments due to the use of thin films (30-50 µm) and the use a 10x objective lens with a low numerical aperture, which gives a near cylindrical geometry over a distance of 30-50 µm in the z-direct." It is surprising to me that a 10x objective gives such a large Rayleigh range in the beam waist. I would have expected the beam to diverge more quickly than this beyond the focal waist and for the shape to not be of simple cylindrical geometry. Can the authors back up this claim with some actual calculations for their particular system?

*[A9]* Based on the referee's comments it is clear that our discussion on Page 7 has led to some confusion. The author is right that the geometry is not exactly a cylindrical geometry. Nevertheless, based on previous work by Deschout et al. it is close enough to a cylindrical geometry to give accurate diffusion coefficients when combined with Equation 3.  To make this clear this section has been modified to the following:

"Although Eq. 3 was derived with the assumption that the degree of photobleaching is independent of the z-direction (i.e. the depth in the thin film), Deschout et al. have shown that Eq. (3) can be used to extract accurate diffusion coefficients when using a 10x objective lens with a low numerical aperture (0.45) together with thin films (120 µm thick), since this combination gives close to a cylindrical photobleached geometry.  In our work we used lower numerical apertures  (0.3-0.4) and thinner films (30-50 µm) than Deschout et al."

*[10]* Photobleaching recovery: Following on from the last point about the geometry of the bleached volume, it must be presumed that there is no recovery following photobleaching apart from that due to diffusion of organic species back into the detection window of this assumed geometry. It would be helpful to confirm this by carrying out photobleaching recovery experiments at very low relative humidity in sucrose where the diffusion is most likely entirely quenched. Have the authors done this?

*[A10]* Besides diffusion, the only other mechanism of recovery that we can think of is reversible photobleaching (i.e. photoswitching). To address the referee's comments, we have carried out the following experiments: We prepared droplets with sizes between 10-50 µm in diameter containing sucrose, water and trace amounts of dye (conditioned at 60 % RH), and we photobleached the dye uniformly throughout the droplet.  Next, we monitored the integrated fluorescence intensity of the entire droplet as a function of time after photobleaching. Since the photobleaching was performed uniformly on the entire droplet, the dye concentration was uniform throughout the droplet after photobleaching, which eliminated the possibility of diffusion due to concentration gradients. Furthermore, since we monitored the integrated fluorescence intensity of the entire droplet diffusion due to concentration gradients would not be detected.  In these experiments we did see a small recovery (fluorescein =15-40%, Rodamine 6G

= 20% and calcein = 10-20% of the photobleached signal) on a short time scale (recovery time = $\lesssim$ 15 sec, $\lesssim$ 50 sec, and $\lesssim$ 20 sec, for fluorescein, Rodamine 6G and calcein, respectively). We attributed this fast recovery to reversible photobleaching, which has been observed previously. To take this reversible photobleaching into account in the revised manuscript when calculating diffusion coefficients, we only used data recorded 15 sec, 50 sec and 20 sec after photobleaching for fluorescein, Rodamine 6G and calcein, respectively.

The discussion above has been added to the revised manuscript (Section 2.3). Note, in the analysis presented in the original manuscript in several cases we didn't include data recorded during this initial fast recover; hence, in several cases the data in the manuscript has not changed. In cases where the data did chance due to the removal of this fast recovery, the changes were not significant and did not change any of the conclusions in our manuscript.

*[11]* Hydrodynamic radius of fluorescein: How confident can the authors be that the hydrodynamic radius is independent of water activity? This would seem to be quite crucial when considering the plausibility of the Stokes-Einstein equation for reproducing the diffusion constants from measured viscosities. Given that many of the estimated diffusion constants overlap with the measured diffusion constants, it seems possible to me that Stokes-Einstein could even be considered to be a good representation of the diffusion constants of the organic dyes, given all of the uncertainties/errors involved (including the uncertainties in RH and the uncertainties in the viscosity measurements of sucrose solutions).

*[A11]* The difference between the measured diffusion coefficient and the Stokes-Einstein predicted diffusion coefficient at a water activity of 0.38 may be partly due to a decreasing hydrodynamic radius of fluorescein with decreasing water activity (Champion et al., 1997). However, the hydrodynamic radius is not expected to vary by an order of magnitude when the water activity is varied from 0.6 to 0.38. Hence, a change in hydrodynamic radius is not expected to explain the entire difference at a water activity of 0.38. To address the referee's comment this discussion has been added to the revised manuscript (Section 3.1).

*[12]* Given the disparity in sizes of the different fluorescent probes, would the authors expect any of them to be better represented by Stokes-Einstein behavior, once their sizes are compared with that of sucrose?

[revised manuscript text omitted]